# Strategies to Enhance the Solubility and Bioavailability of Tocotrienols Using Self-Emulsifying Drug Delivery System

**DOI:** 10.3390/ph16101403

**Published:** 2023-10-03

**Authors:** Nur-Vaizura Mohamad

**Affiliations:** Centre for Drug and Herbal Development, Faculty of Pharmacy, Universiti Kebangsaan Malaysia, Kuala Lumpur 50300, Malaysia; vaizura@ukm.edu.my; Tel.: +06-039-289-7973

**Keywords:** bioavailability, poor water solubility, tocotrienol, self-emulsifying delivery system

## Abstract

Tocotrienols have higher medicinal value, with multiple sources of evidence showing their biological properties as antioxidant, anti-inflammatory, and osteoprotective compounds. However, tocotrienol bioavailability presents an ongoing challenge in its translation into viable products. This is because tocotrienol oil is known to be a poorly water-soluble compound, making it difficult to be absorbed into the body and resulting in less effectiveness. With the potential and benefits of tocotrienol, new strategies to increase the bioavailability and efficacy of poorly absorbed tocotrienol are required when administered orally. One of the proposed formulation techniques was self-emulsification, which has proven its capacity to improve oral drug delivery of poorly water-soluble drugs by advancing the solubility and bioavailability of these active compounds. This review discusses the updated evidence on the bioavailability of tocotrienols formulated with self-emulsifying drug delivery systems (SEDDSs) from in vivo and human studies. In short, SEDDSs formulation enhances the solubility and passive permeability of tocotrienol, thus improving its oral bioavailability and biological actions. This increases its medicinal and commercial value. Furthermore, the self-emulsifying formulation presents a useful dosage form that is absorbed in vivo independent of dietary fats with consistent and enhanced levels of tocotrienol isomers. Therefore, a lipid-based formulation technique can provide an additional detailed understanding of the oral bioavailability of tocotrienols.

## 1. Introduction

Vitamin E is a fat-soluble antioxidant that can only be produced by plants through photosynthesis, hence it needs to be obtained from foods or supplements, as it plays a crucial role in human nutrition. The distribution of vitamin E in palm oil was reported to be approximately 30% tocopherols and 70% tocotrienols [1]. Tocotrienols have similar structures to tocopherols, with a chromanol head denoted by α, β, γ, or δ, which depends on the position number and methyl groups on the chromanol ring [2]. Tocotrienols differ structurally from tocopherols by the degree of saturation at the side chains, with three double bonds at carbons 3, 7, and 11, whereas tocopherols have saturated phytyl side chains. It was discovered to naturally occur at higher levels in certain cereals and vegetable oils, including palm oil, rice bran oil, barley germ, wheat germ, and annatto [3,4,5,6,7].

Tocotrienols are valuable nutraceuticals due to their numerous pharmacological properties, particularly in preventing or treating non-communicable diseases, including cardiovascular, musculoskeletal, metabolic, and skin disorders, as well as cancers [8]. Tocotrienols have biochemical functions that are directly or indirectly impacted by their antioxidant properties, which prevent the non-enzymatic oxidation of various cell components by molecular oxygen and free radicals. In several studies, tocotrienols have been shown to inhibit free radical production via the induction of enzymes such as superoxide dismutase [9,10] and glutathione peroxidase, which neutralise superoxide radical-produced free radicals [11]. In addition, tocotrienols have been extensively studied for their anti-inflammatory properties, and promising scientific evidence has been presented. In light of the fact that inflammation is a full array of physiological responses to a foreign organism, it has been established that inflammation is a significant element in the progression of numerous chronic diseases and disorders. Tocotrienols have been demonstrated to inhibit the expression of inflammatory mediators such as tumour necrosis factor-alpha [10], interleukin [IL]-1 [12], IL-6 [13], and nitric oxide synthase [14]. Antioxidants protect tissues from damage against reactive oxygen species and other free radicals, indirectly preventing unwanted inflammatory responses from occurring in the first place.

Despite their medicinal value, translating tocotrienols into a viable medicinal product remains challenging due to their low oral bioavailability. Oral administration remains the preferred choice for drug delivery due to its safety, patient compliance, and self-administration capacity. Although oral delivery is the most convenient, it is also limited by a number of barriers in the gastrointestinal tract (GIT) [15]. Upon oral administration, drug solubilisation in the GIT is essential for absorption, as insufficient dissolution can cause incomplete absorption, low bioavailability, and high variability [16]. Tocotrienol has unfavourable physicochemical properties as a highly viscous oil that is nearly insoluble in water and readily oxidised by atmospheric oxygen. Other limitations exist, such as saturable intestinal absorption and the selectivity of the transfer protein [17,18]. These physicochemical, biological, physiological, and anatomical factors act independently and in concern to limit tocotrienol bioavailability and prevent effective oral delivery [19]. Using a rat model, an in vivo study found that intraperitoneal and intramuscular administration of tocotrienols resulted in minimal absorption, whereas oral administration resulted in incomplete absorption of tocotrienols [20]. Researchers have invented various methods to improve the bioavailability of these compounds. One of the innovations is a self-emulsifying drug delivery system (SEDDS), an isotropic mixture of oils, surfactants, co-surfactants, and co-solvents. SEDDSs can improve the oral absorption of highly lipophilic drug compounds by maintaining them in a solubilised state, thereby preventing an improperly water-soluble compound from solubilising and subsequently dissolving [21]. The oral bioavailability of each tocotrienol form was reported low in previous studies, with α-tocotrienol at 27.7%, γ-tocotrienol at 9.1%, and δ-tocotrienol at 8.5% [22]. Therefore, without α-tocopherol, tocotrienol absorption is virtually nonexistent without suitable conditions and optimal fat levels. The findings also showed that the elimination half-lives of several tocotrienol forms ranged between 2.3 and 4.4 h, much shorter than α-tocopherol elimination half-lives, which lasted 48 to 72 h [23,24,25]. The poor and inconsistent oral bioavailability of fat-soluble compounds in the GI led researchers to explore solutions to overcome these issues and ensure positive therapeutic effects in humans.

In this review, the composition and formulation of tocotrienols using SEDDSs, as well as study populations on tocotrienol bioavailability, will be discussed. Tocotrienols have been the subject of a number of human studies over the past decade. However, many factors complicate tocotrienol bioavailability and remain unanswered.

## 2. Pharmacokinetics of Tocotrienol

Tocotrienol exhibits many beneficial effects, but its poor oral bioavailability aids in its application as a future therapeutic agent. Many factors influence tocotrienol pharmacokinetics, such as their solubility, absorption, distribution, and elimination. Absorption is the process by which drugs or substances enter the body. Given any method other than intravenously, drugs or substances molecules must pass tissue barriers such as the skin epithelium, subcutaneous tissue, gut endothelium, and capillary wall to enter the blood. Meanwhile, distribution refers to how drugs or other substances are transported throughout the body. It is then followed by the metabolism or biotransformation of toxicants that function to detoxify, which is divided into two stages: Phase I and Phase II. Phase I includes oxidation, hydrolysis, and reduction mechanisms, where these reactions, catalysed by hepatic enzymes, generally convert foreign compounds into phase II derivatives. Phase I products can be excreted as such if polar solubility permits translocation.

Meanwhile, Phase II involves the conjugation or synthesis reactions. Standard conjugates include glucuronides, acetylation, and glycine combinations [26]. Lastly, the excretion removes metabolic waste from the body, mainly through various processes performed by various body parts and internal organs. The effectiveness of vitamin E in those tissues is significantly influenced by the delivery of orally administered vitamin E to vital organs. Thus, mechanisms underlying the delivery of absorbed vitamin E to tissues have been the subject of active investigation. As the palm vitamin E mixture contains both α-tocopherol and tocotrienols, interactions between these compounds would affect their pharmacokinetics. Of note, α-tocopherol is suggested to affect the bioavailability of tocotrienol significantly. In a study by Qureshi et al., 33 healthy male subjects were supplemented with δ-tocotrienol at 125, 250, or 500 mg/d, dose dependently increasing the plasma area under the curve (AUC). The findings indicated that tocotrienols augmented in the absence of tocopherols, as δ-tocotrienol had better bioavailability, thus enhancing therapeutic properties by reducing the levels of cytokines related to inflammation [23].

### 2.1. Absorption of Tocotrienol

The bioavailability of an orally administered compound comprises three parts products, including the dissolved and the part that escapes degradation in the lumen. Next is the fraction that absorbs and penetrates the entire intestinal membrane and escapes the intestinal first-pass metabolism. The last part is the fraction that escapes the liver’s first-pass metabolism [27]. Tocotrienols are natural lipophilic compounds that exist as oily liquids. Their oral bioavailability was poor and erratic due to low aqueous solubility and miscibility [20]. It is widely believed that all forms of vitamin E are absorbed by passive diffusion [28,29]. This type of diffusion is defined as a substance moving toward the concentration gradient without any energy input from a higher concentration area to a lower concentration area. Thus, the therapeutic level of tocotrienol might be difficult to reach in the blood and target tissue using oral administration [30]. Previous studies have discovered that taking tocotrienols with food increases the amount of tocotrienols absorbed through the intestinal wall. Food like a fatty meal enhances tocotrienol solubility due to the formation of mixed micelles that increase the area of absorption in the intestines caused by stimulating bile salts and pancreatic enzyme secretion [25,31]. Indeed, mixed micelles can solubilise hydrophobic components and diffuse into the unstirred water layer (glycocalix) to approach the brush border membrane of the enterocytes. In addition, food also increases the lymph lipid precursor pool inside the enterocytes, eventually enhancing lymphatic transport. Vitamin E does not stabilise for long in the digestive tract after being ingested, leading to a rapid attack by the digestive enzyme pancreatic lipase before breaking down fat molecules into tiny pieces called fatty acids and mono-glycerides.

According to Fairus et al., tocotrienols disappear rapidly from plasma within 24 h, raising questions about their biological activities [32]. This might be partly due to the low affinity of the α-tocopherol transport protein (α-TTP) for tocotrienols. The findings showed that the repackaging of α-tocopherol in the liver into very low-density lipoprotein cholesterol results in a longer shelf life and higher plasma concentrations of α-tocopherol. A different pathway for tocotrienol absorption has been proposed, which is independent to TTP [33]. In addition, some researchers identified that absorption of γ-tocotrienol was not complete, although an increase in solubility after being administrated with food significantly enhanced bioavailability [20]. Increased levels of γ-tocotrienol in the intestinal lumen cause saturation of tocotrienol absorption, which explains why increasing the dose does not increase tocotrienol bioavailability, indicating barriers and specific mechanisms of oral absorption transport by primary enterocytes. The intestinal uptake of γ-tocotrienol is inversely proportional to its presence in the intestinal lumen, which strongly suggests that the transportation across the intestinal membrane involves a carrier-mediated process [34]. Studies showed that the intestinal permeability of γ-tocotrienol decreased at a concentration of more than 25 µM, indicating that the intestinal uptake of the same was saturable, and a carrier-mediated process was involved. The in situ findings revealed the role of Niemann-Pick C1-like 1 (NPC1L1) as an intestinal transporter in both γ-tocotrienol and α-tocopherol intestinal uptake [34,35]. NPC1L1 is a critical mediator of cholesterol absorption and is mainly found in the GIT epithelial cells and hepatocytes.

### 2.2. Distribution of Tocotrienol

After oral administration, tocotrienol is absorbed from the intestine and transported to the systemic circulation through the lymphatic pathway. It is then incorporated into triglyceride-rich chylomicrons, and some of it is transported to other organs. The distribution of tocotrienol was examined by considering the tolerable upper intake level of vitamin E. The experiment using rats set the limit for tocotrienol daily intake to not exceed an amount in rats equivalent to the acceptable daily intake for humans, approximately 600–800 mg/day [36]. A study by Ikeda et al. reported that two isomers of tocotrienol (α and γ) were distributed in the adipose tissues and skin of the rats fed with palm tocotrienol diet [37]. However, these palm tocotrienols were not detected in the brain, liver, kidney, or plasma. A different source of tocotrienol from rice bran also showed that γ-tocotrienol was predominantly present in the adipose tissue and highly in the skin and heart [38]. Other findings demonstrated that γ-tocotrienol was predominant in adipose tissue on day 3 and was still higher on day 14 after administering a single dose of emulsified γ-tocotrienol subcutaneous injection. This finding aligned with previous studies in which γ -tocotrienol was administered in the diet and showed a preference for accumulation in adipose tissue regardless of the route of administration. Interestingly, γ-tocotrienol levels in the heart and spleen increased significantly on day 14 compared to day 3 [39]. Of note, tocotrienols predominantly accumulated in various sites of white adipose tissues, including epididymal fat, perirenal fat, and visceral fat, as well as in the skin after supplementation with a few concentrations of tocotrienol [40]. Increasing the tocotrienols intake increased its concentration in most tissues, although no dose-dependent effects were observed in the brain site. Despite unclear mechanisms on how tocotrienols selectively accumulate in these tissues, the researchers hypothesise that it depends on the affinity of tocotrienol towards the vitamin E-binding proteins and also due to cytochrome P450 expression level in each organ.

A long-term dietary intake of tocotrienol leads to the accumulation in adipose tissue for 8 weeks [41]. Interestingly, the dietary α-tocopherol reduced the α-tocotrienol but not γ-tocotrienol concentration in various tissues. The findings demonstrated a significant amount of α- and γ-tocotrienol accumulated in the perirenal adipose tissues and epididymal fat of rats fed with a tocotrienol mixture. In contrast, other findings showed that γ-tocopherol concentrations were lower in the adipose tissues than in some tissues, including the adrenal gland, liver, and spleen of rats fed with γ-tocopherol [42]. Although the affinity of α-tocotrienol for α-TTP is nearly identical to γ-tocopherol, it does not explain the tissue-specific accumulation of tocotrienol for α-TTP. Furthermore, it was observed that the concentrations of α- and γ-tocotrienol in perirenal adipose tissue were 9.4- and 36.4-fold greater, respectively, than those in the serum. In contrast, the concentrations of α- and γ-tocopherol were only 0.9- and 1.7-fold higher in adipose tissue compared to serum. The findings suggest that adipose tissue preferentially absorbed or stored tocotrienol than tocopherol.

### 2.3. Metabolism of Tocotrienol

Vitamin E has several forms, containing the same chromanol ring and a hydrophobic side chain 13 carbons long. Tocotrienols differ from tocopherols in that they have an unsaturated side chain that contains three double bonds at the 3′, 7′, and 11 positions, as opposed to tocopherols. Upon supplementation, tocotrienols accumulate only in the skin and adipose tissues, whereas tocopherols can be found in most tissues [43,44]. These findings show that tocotrienols are metabolised and eliminated more extensively and quickly than tocopherols. Once tocotrienols and tocopherols are absorbed and delivered to the liver, their fates will likely undergo metabolism and excretion.

Findings demonstrated that the tocotrienols are metabolised essentially, like tocopherols, as few metabolites degraded from γ-tocotrienol, like carboxyethyl hydroxychroman, carboxymethylbutyl hydroxychroman, carboxymethylhexenyl hydroxychroman, and carboxydimethyloctenyl hydroxychroman, were identified, similar to α-tocopherol [45]. Tocopherols and tocotrienols are metabolised without modifying the chromanol ring through oxidative degradation of the hydrophobic side chain. The mechanism involved was cytochrome P450 catalysed, hydroxylation, and oxidation of the 13′-carbon to form 13′-carboxy chromanol (13′-COOH), followed by a series of stepwise β-oxidations to remove a 2- or 3-carbon moiety from the side chain each cycle [46,47]. Recently, researchers discovered that γ-tocotrienol undergoes metabolism to produce novel metabolites, including sulphated 9′, 11′, and 13′-carboxy chromanol, both in human lung epithelial A549 cells and in rats [48]. In this study, sulfation likely occurs parallel to β-oxidation during tocopherol metabolism, indicating tocotrienol is metabolized much faster and more extensively. Similarly, the plasma concentrations of most metabolites were higher in rats supplemented with γ-tocotrienol than in rats fed with tocopherol [49].

### 2.4. Excretion of Tocotrienol

There are two main routes through which vitamin E is excreted. The most common route of excretion is through bile, which is then excreted in faeces after passing through the liver. Alternatively, vitamin E can be excreted in the urine after chain shortening, similar to β-oxidation, making it more water soluble. Due to a lack of TTP, α-tocopherol is not secreted back into the bloodstream. Therefore, an excess of vitamin E is not accumulated in the liver because it does not accumulate toxic amounts of vitamin E, which leads it to be metabolised and excreted in the bile [50]. A recent investigation revealed that α-tocopherol is found in urine as the metabolite α-carboxyethyl-hydroxychroman (CEHC), where the urinary α-CEHC excretion increased by 0.086 μmol/g creatinine with every 1 mg (2.3-μmol) increase in dietary α-tocopherol [51]. CEHC in urine increased in response to tocotrienol supplements, suggesting that tocotrienols are metabolised in the same way as tocopherols [52]. A study by Lodge et al. found that the γ-CEHC excretion time course increased urinary γ-CEHC at 6 h and a peak at 9 h following ingestion of 125 mg γ-tocotrienyl acetate [52]. However, the percentage of the dose recovered as metabolites in the urine is low (only up to 8%). In addition the long-chain carboxychromanols, especially 13′-COOHs, were found in tissues and faeces in animals supplemented with γ-tocopherol, δ-tocopherol, and γ-tocotrienol [53]. Both tocotrienol and tocopherol are bioavailable in the plasma according to research on vitamin E forms in rats, and they are primarily excreted as unmetabolized forms and long-chain metabolites, including 13′-COOHs in faeces, with more metabolites from tocotrienols than from tocopherols [53].

## 3. Lipid-Based System

Lipid-based drug delivery systems are widely used to enhance the stability of oral active pharmaceutical ingredients. The formulation acts as a solubiliser in the colloidal dispersion to increase drug absorption. Due to their ability to improve oral bioavailability, SEDDSs have gained attention since they allow for a reduction in dosages, a better understanding of drug absorption temporal profiles, targeted delivery of drugs to specific gut absorption windows, and protection against adverse environmental effects. Approximately 60–70% of drug molecules are reported to be insufficiently soluble in aqueous media and have a very low permeability, allowing adequate and reproducible absorption from the GIT following oral administration [54]. The very small droplets of the nanoemulsion facilitate better drug absorption and targeting. It improves the conventional emulsion system and provides new opportunities for more precise design of other drugs, which have better bioavailability and accurate dosage, thereby minimising side effects. SEDDSs are an important method to enhance the bioavailability of hydrophilic compounds. The lipid-based formulation has been classified into four types in relation to their pharmacokinetic properties [55]. Type I consists exclusively of oils, whereas Type II combines oils with water-insoluble surfactants. Type IIIA consists primarily of oils and a small proportion of water-soluble surfactants. Type IIIB, therefore, consists of a small proportion of oils and a majority of water-soluble surfactants and hydrophilic co-solvents. Lastly, Type IV comprises only hydrophilic surfactants and co-solvents [56]. The SEDDS is the Type II lipid based-formulation of interest, making it suitable to deliver compounds classified as Biopharmaceutics Classification System (BCS) Class II, which are poorly water-soluble but have high membrane permeability (Figure 1) [57,58]. While developing an emulsion, drugs crystallise and precipitate, leading to unpredictable pharmacokinetic responses. As a result, liquid SEDDSs can contain less medicine, with significant drawbacks when constructing BCS class II and IV pharmaceuticals [59]. However, by varying the compositions of surfactants and co-surfactants, some medicines, such as insulin [60], cyclosporine [61], ritonavir [62], ibuprofen [63], and calcitriol [61], have been successfully formulated using SEDDSs, which has been used for diabetes, immunosuppressants, HIV antivirals, antipyretics, or analgesic and calcium regulators.

Nanocarriers like solid–lipid nanoparticles (SLNs), nanostructured lipid carriers (NLCs), and polymeric nanoparticles have also been used as vitamin E delivery platforms. SLNs comprise a lipid monolayer surrounding a hydrophobic solid–lipid core, enabling lipid-soluble substances to be incorporated. NLCs, considered the second generation of SLNs, involve the mixture of solid and liquid oil matrices to form a solid colloidal dispersion with particle sizes ranging from 10 to 1000 nm [64]. Mixing lipids with low and high melting points results in irregularities in the crystalline lipid core of NLCs, enhancing their capacity to incorporate compounds [65]. This characteristic offers benefits, especially when dealing with lipophilic compounds such as tocotrienols. A previous study demonstrated that NLCs outperform SLNs in stability and compound loading [66]. Nanoparticles made from polymeric materials consist of amphiphilic polymers with multiple polymer chains exhibiting varying degrees of hydrophobicity. These polymers form self-assembled micelles in an aqueous solution [67]. The active compounds and polymers are dissolved in organic solvents that are immiscible with water, and they are mixed while constantly stirring, resulting in the formation of nanoparticles ranging in size from 10 to 170 nm. Examples include a hybrid system using poly (lactic-co-glycolic) acid (PLGA) and chitosan prepared by synthesizing PLGA-tocotrienol copolymer [68]. Polymers have been demonstrated to provide numerous advantages, especially in improving the solubility and bioavailability of lipophilic substances, such as vitamin E. Meanwhile, SEDDS nanoemulsions represent kinetically stable mixtures of two immiscible phases, aqueous and oil. These formulations yield smaller droplets in the presence of surfactants, leading to faster lipid digestion rates [69].

### 3.1. Components of SEDDSs

Several factors must be considered to create a successful SEDDS with the most significant therapeutic effect. This includes properties of the active moiety and excipients, the potential interactions between drugs and excipients (both in vitro and in vivo), and physiological factors that facilitate or inhibit bioavailability. Regulations, solubility capacity, miscibility, physical state of the excipients at room temperature, digestibility, and compatibility with the capsule shell must also be considered during formulation [70]. This rational approach not only helps to reduce the time involved in formulation development but also helps to reduce the cost of formulation development [71]. SEDDSs are an isotropic mixture of oils and surfactants intermittently accompanied by a co-solvent, which tends to emulsify under mild agitation, mimicking the GIT [72]. When exposed to aqueous media like GI fluids, SEDDSs will self-emulsify to form oil and water nanoemulsions or microemulsions with average droplet sizes between 100 and 300 nm [73,74]. In order to develop suitable SEDDSs, the nanoemulsion areas of mixtures comprising different ratios of surfactants, oils, and co-solvents were determined. Each of the components of SEDDS, i.e., lipids (natural/synthetic origin), surfactants (hydrophilic/hydrophobic), co-surfactants, and co-solvents, have their respective functions to improve the bioavailability of a compound and were well discussed below.

#### 3.1.1. Lipids (Natural/Synthetic)

The lipid or oil component contains different oils with the maximum capacity to solubilise a particular compound. Oil usually has two major positive effects on bioavailability enhancement: drug solubility improvement and a favourable effect on lymphatic transport. However, high solubility may not guarantee high in vivo efficiency and should not be used as the sole parameter for drug optimisation. The lipid or oil component also facilitates lymphatic transport of the drug. In the duodenum, the secretion of cholesterol and bile salts is induced by the presence of lipids and their digested products. They will help the lipid to form micelles [75]. The polar groups of micelles will appear at the aqueous edge, whereas the hydrophobic groups remain at the core. Modified or hydrolysed vegetable oils are one of the best emulsification systems. Both long-chain triglycerides (LCTs) and medium-chain triglycerides (MCTs) can be used with varying saturation levels. MCTs with 6–12 carbon chains are transported through portal blood into the systemic circulation, whereas LCTs with more than 12 carbon chains are transported through the intestinal lymphatics. In general, MCT is preferred over LCT due to its ability to shorten transit time in the GIT while exerting less physiological burden when metabolised [76,77]. It is, therefore, preferred in parenteral drug delivery and formulation since MCT imposes a lower metabolic demand on the body. Compared to LCT, the rate of MCT hydrolysis was significantly higher, demonstrating that MCTs have a faster metabolism than LCTs. This has been evidenced in research where MCTs are quickly transformed into fatty acids in intestinal epithelial cells. These fatty acids are promptly transported into the portal vein, bypassing the lymphatic system and general circulation. Subsequently, they undergo oxidation in the liver and are stored in adipose tissue [78]. The most commonly used oils in the formulation of SEDDSs for MCT and ester-related products were Caprylic/capric triglycerides (Akomed E, Akomed R, Miglyol 810, and Captex 355, Neobee M5^®^ (Stepan Company, Maywood, N.J., USA), Crodamol GTCC^®^ (Croda Inc, Snaith UK)) and fractionated coconut oil (Miglyol 812). While Soybean oil, arachis oil, castor oil, cottonseed oil, maize (corn) oil, and hydrolysed corn were for LCT.

#### 3.1.2. Surfactants (Hydrophilic/Hydrophobic)

The second required component in a SEDDS is a surfactant, amphiphilic molecules composed of polar (hydrophilic) and nonpolar (lipophilic) groups. Based on the hydrophilic-lipophilic balance (HLB) value of the surfactant, it may form oil in water O/W (water is the continuous phase), water in oil W/O (oil is the continuous phase), or a bicontinuous (comprises an equal amount of water and oil) microemulsion. When a surfactant is dispersed in water or an oil and water mixture, it associates with various equilibrium phases in the form of spherical, hexagonal, rod-shaped, and lamellar phase micelles, depending on intermolecular forces. Conversely, reverse micelles can form in a polar liquid such as alkanes if the liquid is a polar [79]. Surfactants increase permeability by partitioning into the cell membrane and disrupting the lipid bilayer’s structural organisation, enhancing permeation. The most commonly used surfactants for the formulation of SEDDSs are water-soluble surfactants like cremophor EL or Tween 80 that function as a promising vehicle for successful drug incorporation depending on drug solubility in the mixture [80]. Above their critical micelle concentration, these materials dissolve in pure water at low concentrations to form micellar solutions. Therefore, it was a proven safety profile rather than their advantages in physicochemical performance. It is critical to accurately determine the surfactant concentration because excessive amounts can cause gastrointestinal irritation. On the other hand, the extremely small lipid droplet size produced by these formulations promotes rapid stomach emptying and wide dispersion throughout the GIT. This reduces exposure to high local surfactant concentrations and, as a result, reduces irritation potential.

Poor water solubility or permeability remains the major challenge for therapeutic drugs to exert maximum effectiveness. The Food and Drug Administration (FDA) has approved D-α-tocopheryl polyethylene glycol succinate (TPGS) as a safe adjuvant in drug delivery systems. TPGS can be used as a solubiliser, absorption enhancer, emulsifier, and surface stabiliser. It has been widely used to fabricate nanodrugs and other formulations, especially BCS class II and IV medications. Low concentrations of TPGS also enhance intestinal lymphatic transport and chylomicron secretion. As a surfactant, TPGS increases drug absorption through different biological barriers. For instance, TPGS was used to fabricate repaglinide nanocrystals that were 25 and 15 times more bioavailable when compared with free drugs [81]. Studies have shown that TPGS enhances colonic drug permeation [82]. The effects of TPGS on icariside II absorption were also examined in a Caco-2 monolayer and four-site rat intestinal perfusion models. It was found that TPGS decreased the efflux ratio and increased the apparent permeability coefficient value of icariside II in Caco-2 monolayers. The hydrophilic shell of TPGS micelles, combined with the lipophilic core, can further enhance the oral bioavailability of transported drugs, acting simultaneously as a nanocarrier and enhancing permeation. Additionally, TPGS micelles prevent drugs from prematurely degrading in the GIT and prolong their residence time and systemic concentration [83]. A study showed that formulations containing TPGS and resveratrol, a natural non-flavonoid polyphenol with diverse biological activities, sustained drug release, and good drug-loading capacities. TPGS, as a host of poorly soluble compounds, ameliorated the water solubility of resveratrol and reduced its intrinsic cytotoxic effects [84]. Resveratrol is effectively loaded in the TPGS nanoemulsions, and their loading efficiencies were found to be 99.38% [85]. Another study demonstrated that all-trans-retinoic acids encapsulated in TPGS had a small diameter range, decreased polydispersity, and negative zeta potential, which improved their solubility [86]. The formulations also enhanced cytotoxic effects on melanoma cells, suggesting they may be a viable alternative to enhance patient compliance and improve therapeutic outcomes.

#### 3.1.3. Co-Surfactants

The production of an optimal SEDDS requires relatively high surfactant concentrations (generally greater than 30% *w*/*w*). This implies that the concentration of surfactants can be reduced by including co-surfactants. The role of the co-surfactant, together with the surfactant, is to reduce the interfacial tension to a negligibly small, even transient negative value. Upon reaching this value, the interface would expand, forming finely dispersed droplets, and then adsorb more surfactants and co-surfactants until their bulk condition is sufficiently depleted to make the interfacial tension positive once more. The microemulsion is created during this procedure, known as spontaneous emulsification. The presence of the co-surfactants decreases the bending stress of the interface. It allows the interfacial film sufficient flexibility to take up the different curvatures required to form nanoemulsions over a wide range of compositions [87]. The presence of co-surfactants like Labrasol (caprylocaproyl polyoxyl-8 glycerides, HLB of 14) and Transcutol HP (diethylene glycol monoethyl ether, HLB of 4.2) displayed an excellent solubilising capacity for the drug. It was also discovered in another study that Labrasol has the ability to increase intestinal drug absorption while also having high tolerance and low toxicity [88].

#### 3.1.4. Co-Solvents

The co-solvent is also another component of SEDDS. In general, co-solvents like ethanol at very low concentrations decrease viscosity and help in the dispersion process by reducing surfactant concentration. It is believed that co-solvents have been added to lipid-based formulations for at least three different reasons. Early cyclosporin products used ethanol to dissolve the drug during manufacture. More commonly, it has been assumed that co-solvents could be included to increase the solvent capacity of the formulation [89]. The third reason for including co-solvents is to aid the dispersion of systems that contain a high proportion of water-soluble surfactants.

### 3.2. Process of SEDDSs

The lipid formulated from SEDDS compounds can be absorbed by the lymphatic system and transported by lipoprotein, bypassing the hepatic first-pass metabolism. After incorporation with lipoprotein, the compound is transported from the intestinal lymph system to the systemic circulation [90]. The long-chain fatty acids of the lipid component in SEDDSs are processed into triglycerides by re-esterification in the small intestine and packaged into chylomicron, followed by exocytosis by secretion into the lymph vessel. The large chylomicron can pass through the lymphatic junction but not the close junction of blood capillaries. Lipidic components of SEDDSs promote drug absorption by stimulating lipoprotein or chylomicron production [91]. The resulting ultrafine emulsion with a wide droplet size range provides a large surface area for interaction with gastrointestinal membranes [92]. In addition, it is known that intraluminal processing of orally administered lipophilic drugs contained in a lipidic vehicle, which results in mixed micelles with bile salts, is essential for their absorption [93]. Notably, the bioactive effects of the various ingredients in SEDDS formulations have significantly improved the oral bioavailability of the loaded drugs. For example, a high surfactant content in a SEDDS formulation facilitates the opening of tight junctions and the fluidity of membranes [94]. As another promising strategy for improving oral delivery of P-gp substrates, intestinal bypass, and hepatic first-pass metabolism, stimulating the intestinal lymphatic pathway and inhibiting intestinal drug efflux pumps such as P-glycoprotein (P-gp) and intestinal cytochrome P450 3A4 (CYP3A4) have been proposed [95].

Drugs must be added to a SEDDS based on certain physicochemical parameters. This includes self-emulsification assessment using visual evaluation to avoid subjective variations and transparency of the resulting micro or nanoemulsion obtained from the reconstitution of self-emulsifying formulations [96]. The size of the globules is also an important factor in self-emulsion performance, which affects the rate and extent of drug release, as well as the absorption [97]. Moreover, zeta potentials are needed to determine the charges of SEDDS oil droplets [98]. Increasing electrostatic repulsive forces between nanoemulsion droplets will prevent coalescence. In contrast, phase separation occurs when electrostatic forces decrease.

## 4. Oral Bioavailability of Tocotrienol with SEDDSs

In oral administration, bioavailability is best defined as the fractional extent to which an active drug dosage is absorbed and made available to systemic circulation after being taken orally. Hence, poor absorption can result in low bioavailability and low bio-efficiency, thereby preventing the optimal benefit from drugs or compounds. The main aim of a drug delivery system is to reduce drug degradation and prevent side effects by improving bioavailability. A drug delivery system is characterised as a drug formulation into a suitable form for absorption. One strategy for overcoming the limitations of tocotrienol oil’s poor water solubility is to transform it from oil into an oil-in-water emulsion, which is more stable. The concentration of three independent variables in the emulsification process was identified using a surfactant, co-solvent, and homogenisation pressure toward two response variables of droplet size and polydispersity index. As illustrated in Figure 2, SEDDSs containing tocotrienols are incorporated in the lipophilic phase via hydrophobic ions paired with appropriate surfactants. The results demonstrated that the optimised nanoemulsion response values for the tocotrienol-rich fraction of red palm oil using 6.09 weight (wt)% mixed surfactant (Tween 80/Span 80 (63:37, wt)) and 20 wt% glycerol as a co-solvent via homogenisation pressure (500 bar) were reported at 119.49 nm of droplet size and 0.286 of polydispersity index. The findings also demonstrated that mixed surfactants performed better than pure surfactants in terms of nanoemulsion storage stability. At the interface between the oil and water phases, small molecule surfactants can pack well with large surfactants due to larger headgroup-size differences [99].

The efficacy of tocotrienol-rich palm oil fractions formulated with SEDDSs has been investigated. For δ-, γ-, and α-tocotrienols, the ratio of logarithmic-transformed AUC0-values of the novel formulation (Tocomin^®^ 50%, palm olein/soybean oil, Labrasol, and Tween 80) was 2.6, 2.9, and 3.0 times greater than the conventional preparation (Tocomin^®^ 10%, palm olein/soybean oil, and trycaprylin). The novel formulation has lower numerical values for the peak plasma time (Tmax) parameters than the conventional preparation, resulting in a faster onset and rate of drug absorption for the former [100]. Moreover, a study by Alayoubi et al. revealed that SEDDSs formulated with Tween 80, Cremophor EL, and 55% of tocotrienol-rich fraction (TRF) enhanced the TRF release into the aqueous phase and increased its solubilisation [101]. In addition, the SEDDS formulation also significantly improved the oral bioavailability of tocotrienol twofold at doses higher than 2.5 mg/kg doses [102]. Previous studies indicated 2.5–4.5 times higher plasma concentration (Cmax) and plasma tocotrienol AUC of the tocotrienols formulated with SEDDSs [103].

A unique formulation of SEDDSs enhanced the in vivo bioavailability of tocotrienol isomers. A summary of tocotrienol formulated with SEDDSs in animal and human studies is summarised in Table 1. Previous studies reported that the mean droplet size, particle size (measurement of homogeneity and width of the size distribution), and zeta potential values of dispersion generated by SEDDS formulations were 211 ± 14 nm, 0.5 ± 0.04, and −25 ± 3, respectively [28]. Similar findings of SEDDS formulations used also showed that the mean droplet size, particle size, and zeta potential were 117 ± 4 nm, 0.5 ± 0.01, and −14 ± 3, respectively [102]. In vivo, a study demonstrated that δ- and γ- tocotrienols isomers loaded in a SEDDS formulation were significantly higher in oral bioavailability at 0.5 and 2.5 mg/kg doses, respectively [28]. The findings illustrated that oral clearances of both tocotrienol isoforms and elimination half-lives were constant as a function of dose. Hence, the researchers hypothesised that a saturable absorption process could explain the nonlinear characteristics. Consistent with the in vivo study, results from the in vitro study showed that the cellular uptake of δ-tocotrienol loaded in SEDDSs at different concentrations was approximately twofold higher when compared with the mixed micelle (commercial capsule) control group. The results also revealed that the SEDDS formulation enhanced the passive uptake of δ- and γ-tocotrienol, and the endocytosis process counted for both isomers’ cellular uptake was reported at 20–25%. Interestingly, the cellular uptake of δ- and γ-tocotrienol at high and low concentrations was significantly reduced by the presence of surfactants like Cremophor EL or Labrasol. The findings strongly implied that Cremophor EL and Labrasol formed micelles in the lumen that blocked NPC1L1-mediated tocotrienol absorption across the intestinal membrane after SEDDS digestion, decreasing their cellular uptake and in vivo oral bioavailability and, as a result, causing nonlinear absorption kinetic behaviour of both tocotrienol isomers. In another study using a similar formulation of SEDDSs, both in vitro and in vivo oral studies showed a twofold increase in the cellular uptake and oral bioavailability of δ- and γ-tocotrienol incorporated in SEDDSs compared to the commercialized soft gelatin capsule (ToCOVID^®^) [102]. The authors postulated that the SEDDS formulation components enhanced γ-tocotrienol intestinal permeability by acting as permeation enhancers. Recent findings revealed that formulated tocotrienol with SEDDSs improved the plasma tocotrienol fourfold compared with unformulated tocotrienol in osteoporotic rats. In addition, using SEDDS also enhances the effects of tocotrienol on bone parameters, suggesting an increase in the benefit of the tocotrienol when formulated [103].

A number of clinical trials were conducted to examine the bioavailability of tocotrienol and its efficacy in different populations. The bioavailability of tocotrienol isomers under fed and fasted conditions in eight healthy volunteers was investigated. The findings indicated that food was found to increase the onset as well as the extent of absorption of all tocotrienols isomers by more than twofold. In the fed state, the mean apparent volume of distribution values was notably smaller than those in the fasted state. This difference could be attributed to the enhanced absorption of tocotrienols when taken with food [25]. A consecutive study by Yap et al. in a three-way crossover design involving six healthy volunteers showed that formulated tocotrienol with self-emulsifying achieved a faster onset of absorption by a two- to three-fold increase in the extent of tocotrienol bioavailability compared to the non-self-emulsifying oily solution under fasted conditions [104]. It was reported that a shorter lag time was observed by self-emulsifying systems, with approximately one hour, attributed to their ability to form a readily absorbable substance without bile secretion. A different study by Khosla et al. investigated the postabsorptive fate of tocotrienol isomers and their associations with lipoprotein subfractions in humans. The findings demonstrated that in all four young, healthy Caucasians, maximal α-tocotrienol levels were reported at an average of almost 3 μM in blood plasma, 1.7 μM in low-density lipoprotein, 0.9 μM in triglyceride-rich lipoproteins, and 0.5 μM in high-density lipoprotein. The results indicated that the dietary α-tocotrienol was rapidly delivered to the lipoprotein subfractions of human plasma at sufficient concentrations to exert its function [105]. In a randomized, placebo-controlled, blinded endpoints clinical study, the effects of tocotrienol formulated with self-emulsifying for 2 months on arterial compliance and vitamin E blood levels were studied [106]. Significantly higher plasma levels for α-, δ-, and δ-tocotrienol concentrations were reported compared to placebo. In addition, at 100 and 200 mg doses, tocotrienol significantly improved with pulse wave velocity reductions of 0.77 m/s and 0.65 m/s, respectively. However, no significant changes were observed in serum lipid parameters. Still, there was a trend towards improvement in arterial compliance after 2 months of formulated tocotrienol treatment.

Patel et al. found that tocotrienol capsule (ToCOVID) supplementation significantly increased the tocotrienol tissue concentrations in various sites, including the blood, skin, adipose tissue, brain, cardiac muscle, and liver, over time. A suitable concentration of tocotrienol was reported to be neuroprotective in experimental stroke models and lower the model’s end-stage liver disease score by 50%. This evidence demonstrates that orally supplemented tocotrienol for even the shortest duration is transported to the vital organs of adult humans at a detectable level in tissue. The findings provide insight to identify specific mechanisms involved in tissue delivery and metabolism in future studies [107]. In addition, another study using hypercholesterolemic subjects (*n*- = 32) was randomly assigned to orally receive 300 mg of tocotrienols capsules (ToCOVID) daily for 6 months. Supplementation increased tocotrienol concentrations 22-fold from baseline in comparison with the placebo group. A significant reduction in total serum cholesterol and low-density lipoprotein cholesterol decreased significantly by −8.9 ± 0.9% and −12.8 ± 2.6%, respectively, following 4 months of tocotrienol supplementation. The reduction persisted until the end of the 6-month study [108].

Mixed tocotrienol, therefore, was found to show hepatoprotective effects in the first clinical trial involving hypercholesterolemic adults with non-alcoholic fatty liver disease during the 1-year treatment [109]. No adverse reactions with well-tolerated were also observed with tocotrienol supplementation. Following the study, high levels of tocotrienols were measured in all tocotrienols group volunteers, confirming a high level of compliance. Due to favourable effects on animal and human immune systems, the assessment of tocotrienol-rich fraction supplementation on immune response following the tetanus toxoid (TT) vaccine in healthy female volunteers was conducted [110]. The findings showed that tocotrienol supplementation significantly increased the total vitamin E level in the plasma compared with the placebo group, indicating overall compliance. In addition, a significantly enhanced production of interferon-γ and IL-4 by the mitogen or TT-stimulated leukocytes with anti-TT IgG production was observed. The tocotrienol supplementation groups also significantly lowered the IL-6 level compared with the control group.

Available data show that 40–70% of all drug molecules either have extremely low permeability or are insufficiently soluble in aqueous media, preventing them from being sufficiently and consistently absorbed from the GIT after oral administration [111,112]. Very small droplets of the nanoemulsion favour better drug absorption and targeting. It improves the performance of conventional emulsion systems and creates new opportunities for the precise design of other drugs with increased bioavailability and precise dosing, resulting in fewer side effects. Research suggests that tocotrienols have an overall positive impact on human health. The poor absorption of tocotrienol can result in low bioavailability and, subsequently, low bio-efficiency, thereby preventing the optimal benefit. The main aim of the drug delivery system is to reduce drug degradation and prevent side effects by improving bioavailability. Hence, oral bioavailability increases the fraction of an active drug dosage absorbed and available to the systemic circulation after oral administration. The evidence showed that tocotrienols can be detected in plasma after both short- and long-term supplementation, where increases in plasma levels were observed with the SEDDS formulation. Variations in tocotrienol dosage during formulation, as the suggested tocotrienol is still uncertain, may also influence the findings. Nevertheless, data also demonstrated that the formulation has beneficial effects on lipid profile, which warrants further investigation.

**Table 1 pharmaceuticals-16-01403-t001:** Summary of tocotrienol formulation using SEDDSs.

In Vitro/In Vivo/ Human Study	Treatment	Formulation of SEDDSs	Mode/Treatment Duration	Findings	Reference
Male Sprague–Dawley rats (*n* = 18, weight 250–350 g)	δ-T3 and γ-T3 formulated with SEDDS Doses: 0.5, 2.5, and 25 mg/kg	Primary surfactant: Cremophor EL (40.7% *w*/*w*) Co-surfactant: Labrasol (40.7% *w*/*w*) Secondary oil: Captex 355 (7.2% *w*/*w*) Co-solvent: ethanol (11.4% *w*/*w*)	Oral gavage— 45 min (blood samples were collected at 1, 2, 3, 4, 6, 8, 10, and 12 h)	↑ Oral bioavailability— (i) δ-T3 at 0.5 and 2.5 mg/kg doses 0.05 (ii) γ-T3 at 2.5 mg/kg ↑ Passive permeability of δ-T3 and γ-T3 (threefold)	[28]
Caco2 cells	Doses: (i) 1–25 μM for δ-T3 (ii) 0.1–2.5 μM for γ-T3		Incubate—45 min	↑ Cellular uptake at high and low concentrations of δ-T3 ↑ Cellular uptake at 0.1–2.5 μM of γ-T3	
Male Sprague–Dawley rats (*n* = 25, weight 250–400 g)	γ-T3 formulated with SEDDS Doses: 1, 2.5, 10, 25 and 50 mg/kg	Primary surfactant: Cremophor EL (40.7% *w*/*w*) Co-surfactant: labrasol (40.7% *w*/*w*) Secondary oil: captex 355 (7.2% *w*/*w*) Co-solvent: ethanol (11.4% *w*/*w*)	Oral gavage— 45 min (blood samples were collected at 1, 2, 3, 4, 6, 8, 10, and 12 h)	↑ Oral bioavailability: 10, 25, and 50 mg/kg (twofold)	[102]
Caco2 cells	Doses: 0, 5, 10, 15, 20, 25, 30, 35, 40, 45, and 50 μM		Incubate—45 min	↑ Cellular uptake	
Osteoporotic female Sprague–Dawley rats (*n* = 36, weight 200–250 g)	Annatto-T3 formulated with SEDDS Dose: 60 mg/kg	Primary surfactant: Cremophor EL (40.7% *w*/*w*) Co-surfactant: Labrasol (40.7% *w*/*w*) Secondary oil: Captex 355 (7.2% *w*/*w*) Co-solvent: ethanol (11.4% *w*/*w*)	Oral gavage— 2 months treatment	↑ Plasma annatto concentration ↑ Bone parameters (cortical bone thickness, preserved bone calcium content, bone biomechanical strength, and antioxidant enzyme activities)	[103]
Healthy adult male volunteers (*n* = 6, aged 26–41 years old and body weight 55–75 kg)	Tocomin^®^ 50% (21.6: γ-, 6.4: δ-, 10.7 α-tocotrienol and 10.9% α-tocopherol) formulated with SEF Dose: 200 mg mixed tocotrienols at 9:00 a.m. after a 12 h fast with 240 mL of water	SES-A- Surfactants: i. Tween 80 (12.5 mg) ii. Labrasol (87.5 mg) Soybean oil: 351.3 mg SES-B- Surfactants: i.Tween 80 (367.5 mg) ii. Labrasol (52.5 mg) Soybean oil: 31.3 mg	Orally— Blood samples were collected at 0 (before dosing), 1–8, 10, 12, 14, 18, and 24 h after supplemented	Both SES-A and B- ↑ plasma levels and faster rate of drug absorption	[104]
Young healthy Caucasian Women (*n* = 8, aged 23.5 ± 2.2 years old and body weight 58 ± 7.5 kg)	ToCOVID Suprabio^®^ by Carotech Inc, New Jersey contains: 77 mg α-T3 96 mg δ-T3 3 mg γ-T3 62 mg α-TCP 96 mg γ-TCP Dose: 400 mg (one time) with a fat-loaded meal	Tocomin^®^ 50: 200 mg Soya Oil: 305.4 mg Labrasol^®^: 50 mg Cremophor EL: 50 mg	Orally— Blood samples were collected at 2, 4, 6, and 8 h after supplemented.	Tocotrienols were detected in the blood plasma, and all lipoprotein subfractions studied postprandially	[105]
Healthy participants (*n* = 16, age 21–40 years old)	ToCOVID Suprabio^®^ b.i.d. by Carotech Inc, New Jersey contains: 61.52 mg α-T3 112.8 mg γ-T3 25.68 mg δ-T3 Dose: 400 mg (twice 200 mg per day)	Tocomin^®^ 50: 200 mg Soya Oil: 305.4 mg Labrasol^®^: 50 mg Cremophor EL: 50 mg	Orally— Blood samples were collected at 0, 6, and 12 weeks.	↑ Tissue concentrations in blood, skin, adipose, brain, cardiac muscle, and liver ↓ Model for end-stage liver disease score in 50%	[107]
Healthy adult male volunteers (*n* = 8, age 22 ± 47 years and body weight 50 ± 79 kg)	ToCOVID Suprabio^®^ by Hovid Pte. Ltd. Malaysia contains: 21.8 mg α-T3 41.6 mg γ-T3 10.7 mg δ-T3 34.8 int. units d-α-TCP Dose: 300 mg after fasting for a minimum of 12 h overnight and standard meals were given at 4 and 10 h after administration	Tocomin^®^ 50: 200 mg Soya Oil: 305.4 mg Labrasol^®^: 50 mg Cremophor EL: 50 mg	Orally— Blood samples were collected at 1, 2, 3, 4, 5, 6, 7, 8, 10, 14, 18, and 24 h after supplemented	↑ The onset and extent of absorption for all T3 isomers by more than two folds between fed and fasted. ↔ Peak plasma concentration of T3 isomers between fed and fasted Elimination half-life of T3 were 4.5- to 8.7-fold shorter than α-TCP ↔ For elimination half time between fed and fasted	[25]
Healthy male subjects (*n* = 36, aged < 40 years old)	ToCOVID Suprabio^®^ by Hovid Sdn Bhd, Malaysia contains: 23.54% α-T3 43.16% γ-T3 9.83% δ-T3 23.5% α-TCP Doses: 50 mg, 100 mg and 200 mg	Tocomin^®^ 50: 200 mg Soya Oil: 305.4 mg Labrasol^®^: 50 mg Cremophor EL: 50 mg	Orally— Blood samples were collected after 2 months supplemented	↑ Plasma δ, α, and γ- T3 concentrations Linear dose–concentration relationship for all the isomers ↔ On BP and serum TC and LDL-C	[106]
Hypercholesterolemic but otherwise healthy subjects (*n* = 32 (20 males and 12 females), aged between 31 and 53 years old)	ToCOVID Suprabio^®^ by Hovid Sdn Bhd, Malaysia One capsule contains 50 mg of mixed T3: 30.8% α-T3 56.4% γ-T3 12.8% δ-T3 22.9 IU α-TCP Dose: 300 mg daily = 3x after breakfast and 3x after dinner	Tocomin^®^ 50: 200 mg Soya Oil: 305.4 mg Labrasol^®^: 50 mg Cremophor EL: 50 mg	Orally— Blood samples were collected twice at 2 weeks for 24 weeks	↑ Serum T3 concentration relative to the TCP level T3 ↑ 22-fold compared to baseline T3 ↓ TC and LDL at 4 months until 6 months compared with baseline	[108]
Untreated hypercholesterolaemic adults (*n* = 87, aged > 35 years old)	ToCOVID Suprabio^®^ by Hovid Sdn Bhd, Malaysia contains: 61.5 mg α-T3 112.8 mg γ-T3 25.7 mg δ-T3 61.1 mg α-TCP Dose: 400 mg (twice 200 mg per day)	Tocomin^®^ 50: 200 mg Soya Oil: 305.4 mg Labrasol^®^: 50 mg Cremophor EL: 50 mg	Orally— Blood samples were collected after an overnight fast of 1 year treatment period	Normalisation of the hepatic echogenic response in NAFLD	[109]
Healthy women (*n* = 108, aged 18–25 years old)	ToCOVID Suprabio^®^ by Hovid Sdn Bhd, Malaysia contains: 61.52 mg α-T3 112.8 mg γ-T3 25.68 mg δ-T3 91.60 IU α-TCP Dose: 400 mg (twice 200 mg per day)	Tocomin^®^ 50: 200 mg Soya Oil: 305.4 mg Labrasol^®^: 50 mg Cremophor EL: 50 mg	Orally— Blood samples were collected at 0, 28, and 56 days after supplemented	↑ Total vitamin E level in the plasma ↑ Interferon-γ and IL-4 and anti-TT IgG ↓ IL-6 level	[110]

Abbreviation: anti-TT IgG: anti-tissue transglutaminase immunoglobulin G, α-TCP: alpha-tocopherol, α-T3: alpha-tocotrienol, BP; blood pressure, δ-T3: delta-tocotrienol, γ-T3: gamma-tocotrienol, IL-4: interleukin-4, IL-6: interleukin-6, IU: international unit, LDL-C: low-density lipoprotein-cholesterol, mg/kg: milligram per kilogram, NAAFLD: Non-alcoholic fatty liver disease, μM: micromole, SES: self-emulsifying system, SEDDS: self-emulsifying drug delivery system, TC: total cholesterol, ↑: increase, ↓: decrease, ↔: no changes.

## 5. Limitation

SEDDSs are associated with many advantages, but they have fewer limitations, nonetheless. Regarding safety after administration, there is a critical need for consideration when using lipids and surfactants as SEDDS components. Due to the complexities of their properties, these molecules can produce a complex reaction or interaction with a physiological environment that is challenging to monitor and control in vivo. A very high concentration of surfactants, mainly ionic surfactants, may cause severe toxicity by changing the structures of proteins, impairing enzymes, and disrupting phospholipid membranes [113]. When dispersed on a nanoscale, many oil components or lipid compounds can change into toxic materials. Therefore, the Code of Federal Regulation (CFR) has listed materials that the United States FDA has deemed generally safe (GRAS), where the components can be chosen as a guide [114]. However, pharmacokinetics studies are also hampered by the in vivo complexities of the reaction or interaction and the effect, particularly in human volunteers, which need to be considered. Despite its promising effects, only a few studies have reported on tocotrienol bioavailability. The toxicity profiles of tocotrienol nanoformulations were also unclear. To provide significant insights into various applications, toxicities, and pharmacokinetics, more in vivo research and human studies are needed.

## 6. Future Perspective

In the GI system, fats and oils are absorbed by solubilising them into micelles after being secreted into bile salts and acids, followed by lipase digestion. This absorption process has high inter-individual variability, resulting in varying doses between different persons. Bile salts and acid secretions are dependent on and stimulated by a fatty diet that can vary greatly between meals consumed. Similar to other fat-soluble nutrients and dietary lipids, the oral absorption of tocotrienols is characterised by low absorption rates, significant variability, and reliance on various formulation factors. SEEDS are formulated with different types of surfactants and oil carriers to overcome these variables. This formulation approach has empowered SEEDS to consistently achieve enhanced absorption of fat-based vitamins and drugs, including tocotrienols. Lipid-based drug delivery systems enhance transcellular and paracellular drug transport by momentarily disrupting lipid bilayer cells and altering the tight junction by-products of lipid digestion. Interestingly, due to their lipophilic natures, they could pass through complex physiological barriers like the blood–brain barrier without surface modifications [115]. They are also promising carriers for shielding therapeutic peptides from hostile GI conditions [116]. SEDDSs are also promising oral protein therapeutic and genetic material delivery systems. The inclusion of these macromolecules in nanoscale emulsion droplets ensures effective delivery. In addition, new strategies like cell-penetrating peptides (CPPs) have emerged as promising options for next-generation peptide-based drug delivery vectors due to their ability to transport substances across cell membranes with moderate toxicity [117]. However, this role is still in its early stages. Drug delivery can be challenging due to the intestinal mucosal barrier. Nanoemulsions with negative zeta potentials can penetrate the mucus layer, but those with positive zeta potentials more easily absorb the cells. Capricious zeta potentials allow nanoemulsions to penetrate effectively and to be highly absorbed by cells. Therefore, dual-acting zeta-potential micelles were designed to achieve optimal muco-permeation and cellular uptake. Through slow and time-dependent changes in zeta potentials, micellar droplets provide optimal mucus permeation and significant cellular uptake [118].

## 7. Conclusions

Recent reports on the use of SEDDS-formulated tocotrienol are summarised in this review. SEDDSs are simply oily drug solutions that can be absorbed into the body when administered as oily droplets. SEDDSs oily droplets are among the highest mucus-permeating nanocarrier systems. Due to the incorporation of surfactants exhibiting polyethylene glycol-substructures, these mucus-permeating properties can be further improved. Optimisation of SEDDS components, such as oils, surfactants, co-surfactants, and co-solvents, is necessary to enhance the pharmacokinetics of tocotrienol. These promising findings suggest that formulations should be applied as carriers of tocotrienols to achieve better therapeutic applications. In line with the market trend, demand for natural ingredients and new delivery mechanisms is shifting. Tocotrienols are natural nutraceutical ingredients that have a wide array of health benefits. More work exploring various delivery systems will improve tocotrienol bioavailability. The clinical evidence may serve as scientific fundamentals towards better acceptance among consumers, clinicians, and health authorities. Therefore, tocotrienol bioavailability should be reexamined to strategize future research. 

## Figures and Tables

**Figure 1 pharmaceuticals-16-01403-f001:**
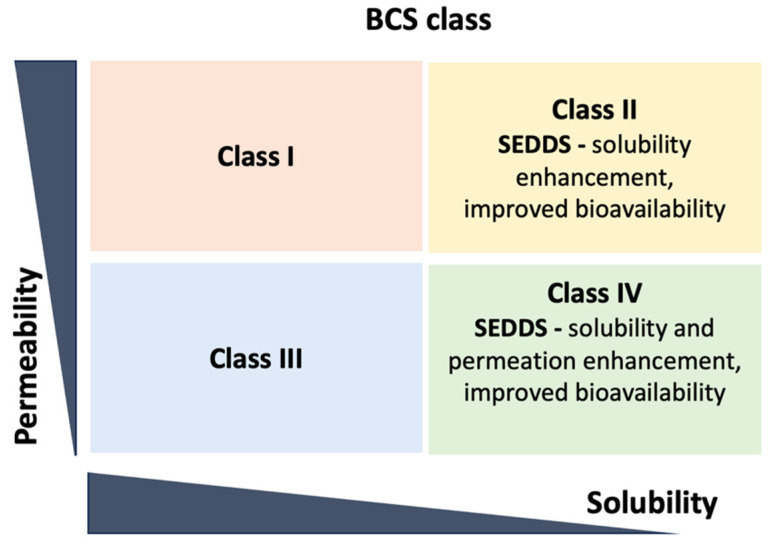
BCS classification categories based on permeability and solubility.

**Figure 2 pharmaceuticals-16-01403-f002:**
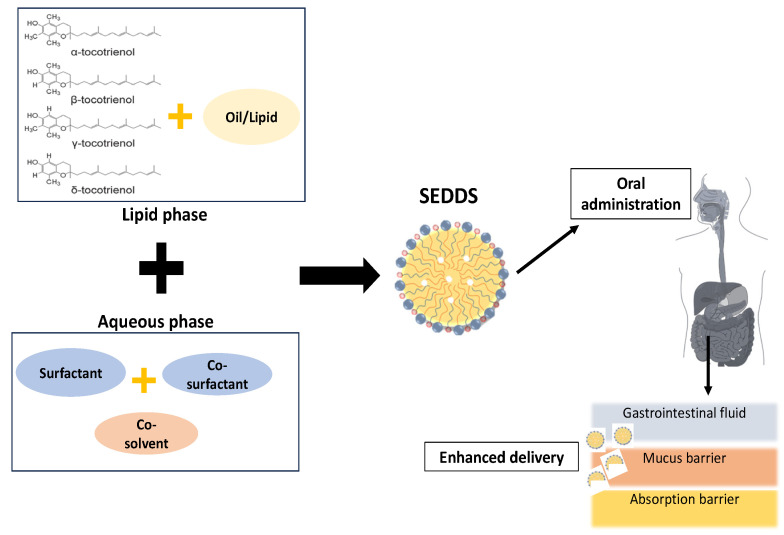
Formulation of tocotrienols with SEDDSs.

## Data Availability

The data used in this study are available within the article.

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
