# Peer review of "Strategies to Enhance the Solubility and Bioavailability of Tocotrienols Using Self-Emulsifying Drug Delivery System"

_pharmaceuticals, 2023, doi:10.3390/ph16101403_

Round 1

Reviewer 1 Report

Article is well drafted. But needed some information on Vitamin E derivative such as TPGS. Just highlight same in article. TPGS also used in various drug delivery systems such as SEDDS and etc. 

NA

Author Response

Thank you for reviewing our manuscript. Your comments are much appreciated and they are responded as the following with changes highlighted in the revised manuscript:

Reviewer 1

Comment 1: Article is well drafted. But needed some information on Vitamin E derivative such as TPGS. Just highlight same in article. TPGS also used in various drug delivery systems such as SEDDS and etc.

Reply 1: Thank you for your comment. I already amended the section highlighting TPGS’s use in drug delivery systems.

“Poor water solubility or permeability remains the major challenge for therapeutic drugs to exert maximum effectiveness. The Food and Drug Administration (FDA) has approved D-α-tocopheryl polyethylene glycol succinate (TPGS) as a safe adjuvant in drug delivery systems. TPGS can be used as a solubiliser, absorption enhancer, emulsifier, and surface stabiliser. It has been widely used to fabricate nanodrugs and other formulations, especially BCS class II and IV medications. Low concentrations of TPGS also enhance intestinal lymphatic transport and chylomicron secretion. As a surfactant, TPGS increases drug absorption through different biological barriers. For instance, TPGS was used to fabricate repaglinide nanocrystals that were 25 and 15 times more bioavailable when compared with free drugs [77]. Studies have shown that TPGS enhances colonic drug permeation [78]. The effects of TPGS on icariside II absorption were also examined in Caco-2 monolayer and four-site rat intestinal perfusion models. It was found that TPGS decreased the efflux ratio and increased the apparent permeability coefficient value of icariside II in Caco-2 monolayers.”

Remarks: Thank you for reviewing the manuscript again. Look forward to receiving a favourable reply from you.

Reviewer 2 Report

This review deals with pharmacokinetic of tocotrienols when they are orally administrated using SEDDS. Oral delivery is not obvious in the title of the review which has to be changed. 

Oral delivery systems usind SEDDS must also be included in introduction. Is there any study concerning this issue (SEDDS oral delivery) conducted? 

Introduction must also include a review (bilbiography) concerning tocotrienols delivery systems (apart from SEDDS), their advantages and their faillures concerning SEDDS.

In Section 2 there is an extentend paragraph (lines 75-89) concerning drug movement (word movement has to be replaced) generally. This paragraph has to be revised. 

Concerning Section 4. At this section author presents results of oral delivery of comercially or lab made tocotrienols delivery systmes (having different formulations of SEDDS). There is a comparison to the concentration of tocotrienols alone found in plasma or in other organs such as skin, bone or brain. Is this concentration directly related to initially administrated tocotrienols? This has to be mentioned. In this section it will be better to categorize the systems concerning the formulation of SEDDS used. Afterwards comparison has to made concerning the initial concerntration and the solubility of tocotrienols in each substabce. These two factors are crucial for the delivery of drugs.

English grammar is used well, hence more scientific expretions has to be used. For example phrases such as "the current review endeavors to provide" (line 70), "The study of drug movement within, through, and out of the body" (line 74) and "in another study" (line 331) has to be replaced.

Author Response

Thank you for reviewing our manuscript. Your comments are much appreciated and they are responded as the following with changes highlighted in the revised manuscript:

Reviewer 2 

Comment 1: This review deals with pharmacokinetics of tocotrienols when they are orally administrated using SEDDS. Oral delivery is not obvious in the title of the review which has to be changed.

Reply 1: Thank you for your suggestion. I already adjusted the title as suggested.

“Strategies to Enhance the Solubility and Bioavailability of Tocotri-enols Using Self-Emulsifying Drug Delivery System.”

Comment 2: Oral delivery systems using SEDDS must also be included in introduction. Is there any study concerning this issue (SEDDS oral delivery) conducted?

Reply 2: Thank you for your comment. I already amended the following suggestion in the introduction.

“Oral administration remains the preferred choice for drug delivery due to its safety, patient compliance, and self-administration capacity. In addition to being the most convenient route of administration, oral delivery has been limited owing to the numerous barriers in the gastrointestinal tract (GIT) [15]. Upon oral administration, drug solubilisation in the GIT is essential for absorption, as insufficient dissolution can cause incomplete absorption, low bioavailability, and high variability [16]...”

“Using a rat model, an in vivo study found that intraperitoneal and intramuscular ad-ministration of tocotrienols resulted in minimal absorption, while oral administration resulted in incomplete absorption of tocotrienols [20]…”

“The oral bioavailability of each tocotrienol form was reported low in previous studies, with α-tocotrienol at 27.7%, γ-tocotrienol at 9.1%, and δ-tocotrienol at 8.5% [22]. Therefore, without α-tocopherol, tocotrienol absorption is virtually nonexistent without suitable conditions and optimal fat levels. Findings also showed that the elimination half-lives of several tocotrienol forms ranged between 2.3 and 4.4 hours, much shorter than α-tocopherol elimination half-lives, which lasted 48 to 72 hours [23-25]. The poor and inconsistent oral bioavailability of fat-soluble compounds in the GI led researchers to explore solutions to overcome these issues and ensure positive therapeutic effects in humans…” 

Comment 3: Introduction must also include a review (bibliography) concerning tocotrienols delivery systems (apart from SEDDS), their advantages and their failures concerning SEDDS.

Reply 3: Thank you for your comment. I have already highlighted the limitations and concerning the tocotrienols delivery system in the limitation section as follows.

“5. Limitation

SEDDS is associated with many advantages, but it has fewer limitations nonetheless. Regarding safety after administration, there is a critical need for consideration when using lipids and surfactants as SEDDS components. Due to the complexity of their properties, these molecules can produce a complex reaction or interaction with a physiological en-vironment that is challenging to monitor and control in vivo. A very high concentration of surfactants, mainly ionic surfactants, may cause severe toxicity by changing the structure of proteins, impairing enzymes, and disrupting phospholipid membranes [104]. When dispersed on a nanoscale, many oil components or lipid compounds can change into toxic material. Therefore, the Code of Federal Regulation (CFR) has listed materials that the United States FDA has deemed generally safe (GRAS), where the components can be chosen as a guide [105]. However, pharmacokinetics studies are also hampered by the in vivo complexity of the reaction or interaction and the effect, particularly in human volunteers, which need to be considered. Despite its promising effects, few studies have reported tocotrienol bioavailability. The toxicity profiles of tocotrienol nanoformulations were also unclear. To provide significant insights into various applications, toxicity, and pharmacokinetics, more in-vivo research and human studies are needed.”

Comment 4: In Section 2 there is an extended paragraph (lines 75-89) concerning drug movement (word movement has to be replaced) generally. This paragraph has to be revised.

Reply 4: Thank you for your suggestion. I have already replaced and revised the following paragraph, as suggested.

“Tocotrienol exhibits many beneficial effects, but its poor oral bioavailability aids in its application as a future therapeutic agent. Many factors influence tocotrienol pharmaco-kinetics, such as their solubility, absorption, distribution, and elimination…”

Comment 5: Concerning Section 4. At this section author presents the results of oral delivery of commercial or lab-made tocotrienols delivery systems (having different formulations of SEDDS). There is a comparison to the concentration of tocotrienols alone found in plasma or in other organs such as skin, bone or brain. Is this concentration directly related to initially administrated tocotrienols? This has to be mentioned. In this section, it will be better to categorize the systems concerning the formulation of SEDDS used. Afterward, comparison has to be made concerning the initial concentration and the solubility of tocotrienols in each substance. These two factors are crucial for the delivery of drugs.

Reply 5: Thank you for your suggestion. In section 4, a discussion on oral bioavailability using tocotrienol formulated with SEDDS has been done. Since there is still limited study conducted using the SEDDS formulation, only 2 different formula of SEDDS has been used which have already been divided into in vivo and human study and further elaborated and discussed in the text. The discussion basically emphasised the plasma level of tocotrienols and their oral bioavailability findings. Therefore, further studies are required to further understand on how the factors might influence the results. In this review, the author only focuses on the SEDDS formulation with current evidence.

Remarks: Thank you for reviewing the manuscript again. Look forward to receiving a favourable reply from you.

Reviewer 3 Report

I have reviewed the article "Self-emulsifying Drug Delivery System of Tocotrienols: Concise Review on Pharmacokinetics". 

The article seems like an introduction to Vitamin E and SEDDS, which is totally in contrast to the title of review. The authors have focused only two very simples topics, like vitamin E and then he has explained SEDDS, its components, which are already known to many readers. To produce novelty,   the authors should write or change this article to " Strategies and pharmaceutical technologies to improve/enhance the solubility and bioavailability of  Tocotrienols" 

The paper must be edited by native English expert as many grammatical mistakes have been found. 

Author Response

Thank you for reviewing our manuscript. Your comments are much appreciated and they are responded as the following with changes highlighted in the revised manuscript:

Reviewer 3

Comment 1: The article seems like an introduction to Vitamin E and SEDDS, which is totally in contrast to the title of the review. The authors have focused only on two very simple topics, like vitamin E and then he has explained SEDDS, its components, which are already known to many readers. To produce novelty, the authors should write or change this article to " Strategies and technologies to improve/enhance the solubility and bioavailability of Tocotrienols"

Reply 1: Thank you for your suggestion. I have already revised the title as suggested.

“Strategies to Enhance the Solubility and Bioavailability of Tocotri-enols Using Self-Emulsifying Drug Delivery System”

Comment 2: The paper must be edited by native English expert as many grammatical mistakes have been found.

Reply 2: Thank you for your suggestion. I already proofread the manuscript with a native English speaker.

Remarks: Thank you for reviewing the manuscript again. Look forward to receiving a favourable reply from you.

Reviewer 4 Report

I reviewed the review entitled "Self-emulsifying Drug Delivery System of Tocotrienols: Concise Review on Pharmacokinetics" submitted by Mohamad NV for publication in Pharmaceuticals. In this manuscript, the author systematically summarized the impact of SEDDS on the improvement of the bioavailability of tocotrienols. The content of this review is well written and summarizes the content and research in a comprehensive way. However, there are few major comments for authors to consider.

1.      In the abstract section, I would suggest that author explain the aim of this review and how does this review differ what is already out there. No need to add the definition of SEDDS in abstract section.

2.      Please explain in the Introduction section in more detail about the research novelty.

3.      Author should add a schematic figure or a graphical abstract that summarizes the basic concept of this review for better understanding of the readers.

4.      Author should add some figures from previously published work. Moreover, author also need to add some independent figures i.e. role of SEDDS to improve the bioavailability of various BCS class drugs.

5.      Elaborate the key parameters for the formation and incorporation of drugs into SEDDS.

6.      In last decade, various strategies of the SEDDS such as zeta potential changing systems, surface modification with cell penetrating peptides etc have been introduced to improve the bioavailability of the loaded drugs. The authors need to add some of these strategies of SEDDS to improve their biomedical applications.  

7.      Conclusion and future perspectives section should described in comprehensive way.

Language needs improvement. The manuscript should be checked for possible typos.

Author Response

Thank you for reviewing our manuscript. Your comments are much appreciated and they are responded as the following with changes highlighted in the revised manuscript:

Reviewer 4

 I reviewed the review entitled "Self-emulsifying Drug Delivery System of Tocotrienols: Concise Review on Pharmacokinetics" submitted by Mohamad NV for publication in Pharmaceuticals. In this manuscript, the author systematically summarized the impact of SEDDS on the improvement of the bioavailability of tocotrienols. The content of this review is well-written and summarizes the content and research in a comprehensive way. However, there are a few major comments for authors to consider.

Comment 1: In the abstract section, I would suggest that the author explain the aim of this review and how does this review differs from what is already out there. No need to add the definition of SEDDS in abstract section.

Reply 1: Thank you for your suggestion. I already removed the sentence as suggested and explained the aim of this manuscript.

“This review discusses the updated evidence on the bioavailability of tocotrienols formulated with self-emulsifying drug delivery systems (SEDDS) from in vivo and human studies. In short, SEDDS formulation enhances the solubility and passive permeability of tocotrienol, thus improving its oral bioavailability and biological actions. This increases its medicinal and commercial value. The self-emulsifying formulation invention also provides a useful dosage form with consistent and enhanced levels of tocotrienol isomers absorbed in vivo upon oral ingestion, independent of dietary fats. Therefore, a lipid-based formulation technique can provide an additional detailed understanding of the oral bioavailability of tocotrienols.”

Comment 2: Please explain in the Introduction section in more detail about the research novelty.

Reply 2: Thank you for your suggestion. I have already explained more detail about the review in introduction section.

Comment 3: Author should add a schematic figure or a graphical abstract that summarizes the basic concept of this review for better understanding of the readers.

Reply 3: Thank you for your comment. I already added the figure summarising this review's basic concept as suggested.

“As illustrated in Figure 2, SEDDS containing tocotrienols are incorporated in the lipophilic phase via hydrophobic ions paired with appropriate surfactants. SEDDS oily droplets are among the highest mucus-permeating nanocarrier systems.”

Figure 2. Formulation of tocotrienols with SEDDS.

Comment 4: Author should add some figures from previously published work. Moreover, author also need to add some independent figures i.e.role of SEDDS to improve the bioavailability of various BCS classdrugs.

Reply 4: Thank you for your comment. I have already amended the suggested figure as in lipid-based system section.

Figure 1. BCS classification categories based on permeability and solubility.

Comment 5: Elaborate the key parameters for the formation and incorporation of drugs into SEDDS.

Reply 5: Thank you for your comment. I have already amended the key parameters for the formation and incorporation of drugs into SEDDS as follows.

“Drugs must be added to a SEDDS based on certain physicochemical parameters. This includes self-emulsification assessment using visual evaluation to avoid subjective vari-ations and transparency of the resulting micro or nanoemulsion obtained from the re-constitution of self-emulsifying formulations [88]. The size of the globules is also an important factor in self-emulsion performance, which affects the rate and extent of drug release, as well as the absorption [89]. Moreover, zeta potentials are needed to determine the charge of SEDDS oil droplets [90]. Increasing electrostatic repulsive forces between nanoemulsion droplets will prevent coalescence. In contrast, phase separation occurs when electrostatic forces decrease.”

Comment 6: In last decade, various strategies of the SEDDS such as zetapotential changing systems, surface modification with cell penetrating peptides etc have been introduced to improve the bioavailability of the loaded drugs. The authors need to add some of these strategies of SEDDS to improve their biomedical applications.

Reply 6: Thank you for your comment. I have already amended the following suggestion in the future perspective section.

“In addition, new strategies like cell-penetrating peptides (CPPs) have emerged as promising options for next-generation peptide-based drug delivery vectors due to their ability to transport substances across cell membranes with moderate toxicity [109]. However, this role is still in its early stages. Overcoming the intestinal mucosal barrier can be a challenge in drug delivery. Those nanoemulsions with negative zeta potentials are effective at permeating the mucus layer, but those with positive zeta potentials are more readily absorbed by cells. Nanoemulsions with capricious zeta potentials can permeate effectively and be highly uptaken by cells. Micelles with dual-acting zeta-potentials in this study were designed to achieve optimal muco-permeation and cellular uptake. Through slow and time-dependent changes in the zeta potential, micellar droplets pro-vide optimal mucus permeation and significant cellular uptake [110].”

Comment 7: The conclusion and future perspectives section should described in a comprehensive way.

Reply 7: Thank you for your suggestion. I have revised the conclusion and future perspective section to be more comprehensive.

“6. Future perspective

In the GI system, fats and oils are absorbed by solubilising them into micelles after being secreted into bile salts and acids, followed by lipase digestion. This absorption process has high inter-individual variability, resulting in varying doses between different persons. The secretion of bile salts and acids is dependent on and stimulated by a fatty diet, which can vary greatly between the different meals consumed. Like all fat-soluble nutrients and dietary lipids, tocotrienol oral absorption is low, highly variable and de-pendent upon formulation parameters. SEEDS are developed using different types of surfactants and oil carriers to circumvent such variables. This has enabled SEEDS to achieve a consistently high fat-based vitamin and drug absorption, including tocotrienols…” 

“7. Conclusions

Recent reports on the use of SEDDS-formulated tocotrienol are summarised in this review. SEDDS are simply oily drug solutions that can be absorbed into the body when administered as oily droplets. As illustrated in Figure 2, SEDDS containing tocotrienols are incorporated in the lipophilic phase via hydrophobic ions paired with appropriate surfactants. SEDDS oily droplets are among the highest mucus-permeating nanocarrier systems. Due to the incorporation of surfactants exhibiting polyethylene glycol- sub-structures, these mucus-permeating properties can be further improved…”

Comment 8: Language needs improvement. The manuscript should be checked for possible typos.

Reply 8: Thank you for your comment. I already proofread the manuscript with a native English speaker.

Remarks: Thank you for reviewing the manuscript again. Look forward to receiving a favourable reply from you.

Reviewer 5 Report

The review gives a superficial information on various aspects of tocotrienol. The author has selected an interesting topic; however, the presentation is poor. The flow of the content is good in the initial phase but it is really tough and confusing in the later stage.  I have a few suggestions to improve this manuscript.

Comments

The content does not match with the title, especially the term-Pharmacokinetics.

There is no deep assessment or discussion on the complete pharmacokinetics of the articles the author has referred.

The author has not included any plasma profiles of tocotrienol in animals or humans, which could have given a better imagination for the readers.

The description in section 3 (Self-emulsifying drug delivery system) needs improvement.

The clinical relevance of Self-emulsifying drug delivery system of tocotrienol needs to be emphasized.

The authors recommendations/comments/suggestions are missing.

Include the structure of tocotrienol.

It is a single author contributed review, hence avoid using “we”…..

Table 1, please mention the references on the right end of the table (after findings).

Author Response

Thank you for reviewing our manuscript. Your comments are much appreciated and they are responded as the following with changes highlighted in the revised manuscript:

 Reviewer 5

I reviewed the review entitled "Self-emulsifying Drug Delivery System of Tocotrienols: Concise Review on Pharmacokinetics" submitted by Mohamad NV for publication in Pharmaceuticals. In this manuscript, the author systematically summarized the impact of SEDDS on the improvement of the bioavailability of tocotrienols. The content of this review is well-written and summarizes the content and research in a comprehensive way. However, there are a few major comments for authors to consider.

Comment 1: There is no deep assessment or discussion on the complete pharmacokinetics of the articles the author has referred. The author has not included any plasma profiles of tocotrienol in animals or humans, which could have given a better imagination for the readers.

Reply 1: Thank you for your suggestion. I already removed the sentence as suggested and explained the aim of this manuscript.

“This review discusses the updated evidence on the bioavailability of tocotrienols formulated with self-emulsifying drug delivery systems (SEDDS) from in vivo and human studies. In short, SEDDS formulation enhances the solubility and passive permeability of tocotrienol, thus improving its oral bioavailability and biological actions. This increases its medicinal and commercial value. The self-emulsifying formulation invention also provides a useful dosage form with consistent and enhanced levels of tocotrienol isomers absorbed in vivo upon oral ingestion, independent of dietary fats. Therefore, a lipid-based formulation technique can provide an additional detailed understanding of the oral bioavailability of tocotrienols.

Comment 2: The description in section 3 (Self-emulsifying drug deliverysystem) needs improvement.

Reply 2: Thank you for your suggestion. I have already explained more detail about the aim of this review in the introduction section.

“…. In this review, the composition and formulation of tocotrienols using SEDDS, as well as study populations on tocotrienol bioavailability, will be discussed. Tocotrienols have been the subject of a number of human studies over the past decade. However, many factors complicate tocotrienol bioavailability and remain unanswered.”

Comment 3: The clinical relevance of Self-emulsifying drug delivery system of tocotrienol needs to be emphasized.

Reply 3: Thank you for your comment. In section 4 most of the studies only showed the oral bioavailability finding with tocotrienols formulated using SEDDS and limited studies have yet been to investigate their clinical relevance.

“The evidence showed that tocotrienols can be detected in plasma after both short and long-term supplementation, where increases in plasma levels were observed with the SEDDS formulation. Varies in tocotrienol dosage during formulation as the suggested tocotrienol is still uncertain may also influence the findings. Nevertheless, data also demonstrated that the formulation has beneficial effects on lipid profile, which warrants further investigation.” 

Comment 4: The authors recommendations/comments/suggestions are missing.

Include the structure of tocotrienol.

Reply 4: Thank you for your comment. I have already amended the following suggestion in the future perspective and conclusion section.

“6. Future perspective

In the GI system, fats and oils are absorbed by solubilising them into micelles after being secreted into bile salts and acids, followed by lipase digestion. This absorption process has high inter-individual variability, resulting in varying doses between different persons. The secretion of bile salts and acids is dependent on and stimulated by a fatty diet, which can vary greatly between the different meals consumed. Like all fat-soluble nutrients and dietary lipids, tocotrienol oral absorption is low, highly variable and de-pendent upon formulation parameters. SEEDS are developed using different types of surfactants and oil carriers to circumvent such variables. This has enabled SEEDS to achieve a consistently high fat-based vitamin and drug absorption, including tocotrienolsFigure 1. BCS classification categories based on permeability and solubility…..”

“7. Conclusions

Recent reports on the use of SEDDS-formulated tocotrienol are summarised in this review. SEDDS are simply oily drug solutions that can be absorbed into the body when administered as oily droplets. As illustrated in Figure 2, SEDDS containing tocotrienols are incorporated in the lipophilic phase via hydrophobic ions paired with appropriate surfactants. SEDDS oily droplets are among the highest mucus-permeating nanocarrier systems. Due to the incorporation of surfactants exhibiting polyethylene glycol- sub-structures, these mucus-permeating properties can be further improved….” 

I have already amended the tocotrienol structure in Figure 2.

Comment 5: It is a single-author-contributed review, hence avoid using“we”

Reply 5: Thank you for your comment. I have already removed the word as suggested.

Comment 6: Table 1, please mention the references on the right end of the table (after findings).

Reply 6: Thank you for your comment. I have already rearranged the reference in Table 1 on the right end of the table after the findings.

Remarks: Thank you for reviewing the manuscript again. Look forward to receiving a favourable reply from you.

Round 2

Reviewer 3 Report

The author must add other technologies and make some comparison with SEDDS for better understanding (to enhance the solubility and bioavailability of of Tocotrienols).  

Author Response

Reviewer 3

Comment: The author must add other technologies and make some comparisons with SEDDS for better understanding (to enhance the solubility and bioavailability of Tocotrienols).  

Reply: Thank you for your suggestion. I already added a few technologies to make the comparison with SEDDS.

“Nanocarriers like solid-lipid nanoparticles (SLNs), nanostructured lipid carriers (NLCs), and polymeric nanoparticles have also been used as vitamin E delivery platforms. SLNs comprise a lipid monolayer surrounding a hydrophobic solid-lipid core, enabling lipid-soluble substances to be incorporated. NLCs, considered the second generation of SLNs, involve the mixture of solid and liquid oil matrices to form a solid colloidal dispersion with particle sizes ranging from 10 to 1000 nm [66]. Mixing lipids with low and high melting points results in irregularities in the crystalline lipid core of NLCs, enhancing their capacity to incorporate compounds [67]. This characteristic offers benefits, especially when dealing with lipophilic compounds such as tocotrienols. A previous study demonstrated that NLCs outperform SLNs in stability and compound loading [68]. Nanoparticles made from polymeric materials consist of amphiphilic polymers with multiple polymer chains exhibiting varying degrees of hydrophobicity. These polymers form self-assembled micelles in an aqueous solution [69]. The active compounds and polymers are dissolved in organic solvents that are immiscible with water, and they are mixed while constantly stirring, resulting in the formation of nanoparticles ranging in size from 10 to 170 nm. Examples include a hybrid system using poly (lactic-co-glycolic) acid (PLGA) and chitosan prepared by synthesizing PLGA-tocotrienol copolymer [70]. Polymers have been demonstrated to provide numerous advantages, especially in improving the solubility and bioavailability of lipophilic substances, such as vitamin E. Meanwhile, SEDDS nanoemulsions represent a kinetically stable mixture of two immiscible phases, aqueous and oil. This formulation yields smaller droplets in the presence of surfactants, leading to a faster lipid digestion rate [71].”

Reviewer 4 Report

Accept in the present form

Author Response

Thank you.

Reviewer 5 Report

The author has responded very well to most of the comments.

Author Response

Thank you.